# Austrian Raw-Milk Hard-Cheese Ripening Involves Successional Dynamics of Non-Inoculated Bacteria and Fungi

**DOI:** 10.3390/foods9121851

**Published:** 2020-12-11

**Authors:** Narciso M. Quijada, Stephan Schmitz-Esser, Benjamin Zwirzitz, Christian Guse, Cameron R. Strachan, Martin Wagner, Stefanie U. Wetzels, Evelyne Selberherr, Monika Dzieciol

**Affiliations:** 1Unit of Food Microbiology, Institute of Food Safety, Food Technology and Veterinary Public Health, Department for Farm Animals and Veterinary Public Health, University of Veterinary Medicine Vienna, Veterinärplatz 1, A-1210 Vienna, Austria; narciso.martin-quijada@vetmeduni.ac.at (N.M.Q.); martin.wagner@vetmeduni.ac.at (M.W.); stefanie.wetzels@vetmeduni.ac.at (S.U.W.); evelyne.selberherr@vetmeduni.ac.at (E.S.); 2Austrian Competence Centre for Feed and Food Quality, Safety and Innovation, FFoQSI GmbH, A-3430 Tulln an der Donau, Austria; benjamin.zwirzitz@vetmeduni.ac.at (B.Z.); cameron.strachan@vetmeduni.ac.at (C.R.S.); 3Department of Animal Science, Iowa State University, Ames, IA 50011, USA; sse@iastate.edu; 4Clinical Unit for Herd Health Management in Ruminants, Department for Farm Animals and Veterinary Public Health, University of Veterinary Medicine Vienna, A-1210 Vienna, Austria; christian.guse@vetmeduni.ac.at

**Keywords:** cheese ripening, bacteria, fungi, quantification, gene-targeted-sequencing

## Abstract

Cheese ripening involves successional changes of the rind microbial composition that harbors a key role on the quality and safety of the final products. In this study, we analyzed the evolution of the rind microbiota (bacteria and fungi) throughout the ripening of Austrian Vorarlberger Bergkäse (VB), an artisanal surface-ripened cheese, by using quantitative and qualitative approaches. The real-time quantitative PCR results revealed that bacteria were more abundant than fungi in VB rinds throughout ripening, although both kingdoms were abundant along the process. The qualitative investigation was performed by high-throughput gene-targeted (amplicon) sequencing. The results showed dynamic changes of the rind microbiota throughout ripening. In the fresh products, VB rinds were dominated by *Staphylococcus equorum* and *Candida*. At early ripening times (14–30 days) *Psychrobacter* and *Debaryomyces* flourished, although their high abundance was limited to these time points. At the latest ripening times (90–160 days), VB rinds were dominated by *S. equorum*, *Brevibacterium*, *Corynebacterium*, and *Scopulariopsis*. Strong correlations were shown for specific bacteria and fungi linked to specific ripening periods. This study deepens our understanding of VB ripening and highlights different bacteria and fungi associated to specific ripening periods which may influence the organoleptic properties of the final products.

## 1. Introduction

Since the first known appearance of cheese production at ~5000 BCE, it has spread worldwide and experienced extensive changes in manufacturing between different regions, affected by distinct technical, social and economic conditions [1,2]. Surface-ripened cheeses (also known as “smear-ripened” or “washed-rind” cheeses due to the application of brine baths during ripening) are produced in many countries worldwide and represent a particular type of cheese that is highly appreciated due to their characteristic taste and flavor that vary widely between the type of cheese and the ripening time and conditions. In surface-ripened cheeses, the activity of the rind microbiota, which differ from the cheese core microbiota, is pivotal for the ripening and the development of the organoleptic properties of the products [3]. Cheese rind microbiota also has a critical impact on product safety, as it acts as a natural barrier against pathogens [4]. The origin of the cheese rind microbiota is uncertain, as some strains can be artificially added to the cheese surface, while others may be present at different surfaces along the cheese manufacture [2,5,6].

Cheese ripening is a complex metabolic process and the cheese rind microbiota undergoes dynamic changes to adapt to nutrient availability, pH, environmental factors (temperature, salt content, etc.) and competition against other microorganisms [7]. The microbial composition of surface-ripened cheeses varies throughout ripening and between the different manufacture conditions and products. Due to the key role of rind microbiota on the production of surface-ripened cheeses, it has been the focus of many studies, where different sequential biochemical and microbial events have been identified [2,5,8,9,10]. At the beginning of the ripening process, the lactic acid, produced after lactose fermentation by the starter cultures (usually strains of *Lactobacillus*, *Lactococcus* and *Streptococcus*), is used by acid-tolerant fungi, such as *Debaryomyces hansenii*. This leads to the deacidification of the cheese surface and to production of secondary metabolites that favor the growth of Gram-positive coagulase-negative cocci (CNC, such as *Staphylococcus*), coryneforms (such as *Brevibacterium* and *Corynebacterium*) and Gram-negative bacteria such as *Halomonas* [4,11,12]. A better understanding of the microbial composition of surface-ripened cheese rinds and their activity would be pivotal for cheesemakers in order to standardize cheese production, enhance the organoleptic properties of the final products and prevent spoilage or undesirable organoleptic properties caused by certain microorganisms [13].

Austrian Vorarlberger Bergkäse (VB) is an artisanal raw milk brine-washed hard-cheese manufactured in the western part of Austria (Vorarlberg) that has a protected designation of origin (PDO) according to the Council Regulation (EEC) of the European Union No. 2081/92. VB are produced without the addition of external ripening cultures and the cheese wheels are brined at different concentration and frequency depending on the product during the ripening, which can last from three to 18 months. The ripening time is a key factor for the organoleptic properties of VB that vary widely between the different times and have a significant impact on the consumer market. Therefore, the evaluation of the microbial events occurring throughout ripening might be fruitful for a deeper understanding of VB ripening towards a more standardized and safer production process while enhancing the characteristic organoleptic properties of this product.

In the last years, our group has studied the microbial composition of VB cheeses throughout ripening and the environmental surfaces present in the ripening cellars by using cloning and Sanger-sequencing, where different *Brevibacterium*, *Corynebacterium*, *Halomonas* and *Staphylococcus* species were found as the most abundant in cheese rinds and on surfaces [14,15]. Additionally, the potential contribution to VB ripening of certain genera often isolated from VB rinds (*Advenella*, *Psychrobacter* and *Psychroflexus*) was investigated by using whole-genome sequencing [16]. More recently, a putative metabolic pathway for histamine degradation has been found in *Brevibacterium* strains isolated from VB rinds [17].

In this study, we aimed to characterize both bacterial and fungal communities from VB rinds at different times throughout ripening (from 0 to 160 days of ripening) by both quantitative and qualitative approaches. Real-time quantitative PCR (qPCR) has previously shown its efficiency for the quantitative investigation of bacterial and fungal communities from dairy products [18,19,20]. Additionally, droplet-digital PCR (dPCR) has arisen as a quantification method that provides more precise estimation of gene copy numbers while not requiring comparison with external standards [21,22,23]. qPCR and dPCR methods were used and compared for fungal quantification in this study by targeting the small subunit (SSU) 18S rRNA gene. For the qualitative investigation of the bacterial or fungal communities in VB rinds, high-throughput near-full-length 16S rRNA or short-read ITS2 gene-targeted sequencing was performed.

## 2. Materials and Methods

### 2.1. VB-Cheese Production

VB-cheese is produced from a single morning milking of raw cow milk, with strict milk production criteria: Milk supply may only use (i) holding without silage production and feeding. Raw milk without thermization, pasteurization, or bactofugation is (ii) partially skimmed (3.3%) and (iii) coagulated with calf-rennet, whey culture and lactic acid autochthonous starter cultures (*Streptococcus thermophilus*, *Lactococcus delbrueckii* spp. *lactis*, and *Lactobacillus casei*). The curd is cut finely and is gradually heated to 51 ${}^{\circ }C$–52.5
${}^{\circ }C$, molded and pressed to eliminate any residual whey. Cheese wheels are soaked in ca. 20% NaCl brine for 2–3 days and then ripened in cellars ( 10 ${}^{\circ }C$–15 ${}^{\circ }C$). Except in facility B (details below), in which dry salting is applied first, cheese wheels are treated with distinct concentration of brine (10–20% NaCl, pH 5.25). No external ripening cultures are added. Typically, the VB-ripening time ranges from three to 18 months. The final cheese wheels weigh between 8 to 35 kg, are 10–12 cm high, and have a diameter of 50–55 cm. Depending on the ripening time, the cheese rind is yellowish-brown to brownish with a hard cheese body texture. Although slightly different dairy technologies and ripening conditions in some facilities are used (Table 1), the VB-cheese is sold as PDO.

### 2.2. Cheese Rind Sampling

Cheese rind samples were taken from two VB-cheese ripening facilities (abbreviated in this study as A and B) located in Vorarlberg, the westernmost federal state of Austria. The facilities A and B process about 7.4 and 1.3 million liters of milk per year, respectively. Samples (n = 200 cheese wheels) were taken from five different ripening stages: directly at the first day of ripening, after the 2–3 days in the brine tank (0) as well as after 14, 30, 90 and 160 days of ripening. For each point in time and ripening facility, 20 samples from different cheese wheels were taken by scraping the entire surface of each cheese with sterile scalpels. Samples were stored at 4 ${}^{\circ }C$ for transportation to the laboratory and processed immediately. Samples were applied to DNA extractions, quantification as well as sequencing approaches.

### 2.3. DNA Extraction

Ten g of cheese rind samples were homogenized in 30 mL sterile Ringer Solution. For subsequent analyses, 250 mg wet-pellet of the homogenized cheese rind sample was applied in duplicate for extraction of genomic DNA (PowerSoil™, MoBio Laboratories, Carlsbad, CA, USA) following the manufacturer’s instructions. Duplicate elutions (250 μL each) were pooled and DNA concentrations were determined with a Qubit^®^ 2.0 Fluorometer (Thermo Fisher Scientific, Vienna, Austria).

### 2.4. qPCR and dPCR Analysis of 18S rRNA Gene

The data for 16S rRNA gene qPCR analysis of total bacterial numbers in VB rind samples (using the same DNA samples) was taken from Schmitz-Esser, et al. [16]. Total fungal DNA was quantified from each of the extracted samples using the 18S rRNA (351 bp) FungiQuant assay [24] (Appendix A—MIQE guidelines qPCR). Each sample was amplified in a qPCR reaction, allowing the number of cycles required for the PCR amplification curve to cross a threshold (Cq) to be calculated for all 200 samples. A single qPCR reaction consisted of 11.95 μL (DEPC)-treated water, 2.5 μL 10 × buffer, 1.75 μL 3.5 mM mgCl2 (stock concentration 50 mM), 0.75 μL of each primer (stock concentration 10 μM), 1 μL of TaqMan^®^ probe (stock concentration 5 μM), 1 μL of dNTP Mix (stock concentration 20 mM, 5 mM of each dATP, dCTP, dGTP and dTTP), 0.3 μL of Platinum Taq DNA polymerase (5 U/μL; Thermo Fisher Scientific) and 5 μL template (genomic DNA). The quantification of DNA was performed in a Mx3000P qPCR instrument (Stratagene, La Jolla, CA, USA) (software v.4.10) after initial denaturation at 94 ${}^{\circ }C$ for 2 min, followed by 45 cycles of 94 ${}^{\circ }C$ for 30 s, 60 ${}^{\circ }C$ for one min. Each reaction was run in duplicate (final volume of 25 μL) using MicroAmp 0.2 mL optical tubes sealed with MicroAmp optical 8-cap strips (Applied Biosystems, Foster City, CA, USA). Additionally, to check for the presence of non-specific products and size of the amplicons, aliquots of qPCR products were verified by agarose gel electrophoresis and by melting curve analysis.

The number of fungal 18S rRNA copies equivalents (FCE) present in a sample was calculated from the Cq by using a standard curve. Standard curves were constructed by using a tenfold dilution series of *Saccharomyces cerevisiae* NCPF 3178 purified genomic yeast DNA. DNA concentration was determined fluorometrically using a Qubit^®^ BR assay (Thermo Fisher Scientific). Copy numbers of the target were calculated using Equation (Equation 1).
(1)$$\mathrm{Number}\;\mathrm{of}\;\mathrm{copies}=\frac{\mathrm{Amount}\;\left(\mathrm{ng}\right)\times 6.022\times 10^{23}}{\mathrm{Length}\;\left(\mathrm{bp}\right)\times 1\times 10^{9}\times 660}$$

The final copy numbers of total fungi (FCE) were calculated using the mean of the copy number per g cheese rinds, including calculation of the DNA volume subjected to qPCR (5 μL), the volume of extracted DNA (2 × 250 μL), and the weight of the sample subjected to DNA extraction (0.5 g cheese rind pellet). An average of 150 18S rRNA gene copies per haploid genome in yeast *Saccharomyces cerevisiae* [25] occurs, therefore, it was taken into account when extrapolating the final fungal copy equivalent (FCE) in this study. All samples were analyzed in duplicates. Negative extraction controls and no-template qPCR controls were included in each qPCR run.

Additionally, to assess the fungal copy number, the NAICA™ SYSTEM for Crystal Digital PCR™ (Stilla Technologies Inc., Villejuif, France) was used. We applied the same 18S rRNA gene qPCR assay [24] for the dPCR by using the same hybridization temperature (60 ${}^{\circ }C$), recommended by Stilla Technologies Inc. and provided a dMIQE guideline experiment summary according to Huggett, et al. [26] (Appendix A—MIQE guidelines dPCR). A set of 200 cheese rind gDNA samples was pooled (n = 10) according to their ripening time points (0, 14, 30, 90 and 160 days). To verify the PCR amplification and avoid underestimation of multicopy target, the DNA was additionally digested by two restriction enzymes: the external enzyme EcoRI (Thermo Fisher Scientific), which cuts DNA at varying distance upstream from the specific target sequence, and the internal enzyme AluI (Thermo Fisher Scientific), which cuts DNA within the target sequence. The protocol for the direct digestion of gDNA for dPCR was taken from international.neb.com [27]. PCRs were carried out using the PerfeCTa Multiplex qPCR ToughMix (Quanta Biosciences, Gaithersburg, MD, USA). Briefly, a single digital PCR reaction consisted of 7.25 μL (DEPC)-treated water, 5 μL PerfeCTa Multiplex, 2.5 μL fluorescein isothiocyanate (FITC) (stock concentration 100 nM) (Thermo Fisher Scientific), 2 μL of each primer (stock concentration 10 μM), 1.25 μL of TaqMan^®^ probe (stock concentration 5 μM), and 5 μL template (genomic pooled DNAs from one ripening point). Samples were loaded on Sapphire chips and amplification was carried out on the Naica Geode machine (Stilla Technologies Inc.), followed by the PCR thermal cycling program: 95 ${}^{\circ }C$ for 10 min, 45 cycles of 95 ${}^{\circ }C$ for 10 s and 60 ${}^{\circ }C$ for 15 s. To avoid saturation of the dPCR, cheese rind DNAs were diluted first to approximately 22 Cq (corresponding to 10${}^{6}$ for qPCR experiment). If the 95% relative uncertainty for the concentration of target molecules (distribution of positive droplet) in the chamber was still too high, the same sample was diluted (1:20) again and reanalyzed.

Image acquisition was performed using the Naica Prism3 reader at 100 ms exposure times for blue channel (FAM). Total droplets enumeration and droplets quality control in the Blue channel was performed by the Crystal Reader software (v2.1.6). Extracted fluorescence values for each droplet were further analyzed using the Crystal Miner software (v2.1.6) (Stilla Technologies Inc., Villejuif, France). The threshold was set automatically at 17,050 by the Crystal Miner software.

The concentration in the final reaction mix for each sample, expressed as stock concentration per μL (Cp/μL), was calculated as previously described [28]. Based on Poisson statistics, the DNA copy numbers per microliter or stock concentration per μL (Cp/μL) was calculated using Equation (Equation 2).
(2)$$\mathrm{Cp}=-\frac{d}{v}\ln\left(1-\frac{p}{n}\right)$$
where *p* is the number of dPCR positive partitions, *n* is the number of total partitions, *v* is the volume of the partition or droplet (Naica System—Sapphire chip–PerfeCTa mix; 0.00058592), *d* represents the combination of the dilution factor used during PCR preparation and for further dilution of the DNA with the dPCR master mix.

### 2.5. qPCR Statistics

The qPCR data (FCE per 0.5 g cheese rind) were analyzed and compared using R (version 3.2.5, *psych* package 1.6.12). The dataset was divided into 10 different subsets based on the two locations (cheese production facilities A, B) and days of ripening (0, 14, 30, 90 and 160). Because the Shapiro-Wilk test showed normal distribution for only two of the 10 subsets, all subsets were described by median and interquartile ranges (IQR). Furthermore, the Wilcoxon Signed-Rank test was used to determine statistical differences (at a significance level p< 0.05) between subsets with the same location based on days of ripening. A *p*-value < 0.05 was considered statistically significant. Owing to the fact that no relationship between the two cheese facilities (A and B) were found (Kendall’s Rank correlation $\tau_{b}$), observed qPCR data from two facilities were analyzed separately.

### 2.6. Gene-Targeted Sequencing

In order to investigate microbial composition in VB rinds throughout time, the DNA purified from the 200 samples described above were pooled regarding their ripening time points (0, 14, 30, 90 and 160 days) and submitted to further gene-targeted amplicon sequencing. With the aim of investigating fungal and bacterial composition, two different sequencing approaches were followed.

Fungal diversity was studied by amplification of the internal transcribed spacer 2 (ITS2) region (size of ca. 300–400 bp), with the primers set ITS3-5′-GCATCGATGAAGAACGCAGC-3′ and ITS4-5′-TCCTCCGCTTATTGATATGC-3′ [29], and sequencing by using a MiSeq Illumina platform, leading to 2,334,863, 250 bp paired-end reads. PCR amplification, sample multiplexing, library preparation and sequencing were performed by Microsynth AG (Balgach, Switzerland).

For examining the bacterial diversity, the 16S rRNA gene was selected for amplification by using the bacteria-specific primers 27F (5′-AGRGTTYGATYMTGGCTCAG-3′) and 1492R (5′-RGYTACCTTGTTACGACTT-3′), and submitted to full-length sequencing by using a PacBio Sequel platform, leading to 224,160 sequences with a median of sequences per sample of 21,117. 16S rRNA gene amplification from total DNA and full-length sequencing by using a PacBio Sequel platform after library preparation was performed by Next Generation Sequencing Facility at the VBCF-Vienna BioCenter (www.viennabiocenter.org, Vienna, Austria).

### 2.7. Bioinformatic Data Analysis

After sequencing, fungal and bacterial amplicon data was analysed independently by following similar approaches. First, quality filtering of the reads, merging of the paired ends (only for ITS2 amplicon sequences), chimera removal and identification of Amplicon Sequence Variants (ASVs) were performed by using *dada2* [30] in R environment [31], following different pipelines for fungal or bacterial data. ASVs rely on single nucleotide differences between sequences and can be considered as Operational Taxonomic Units (OTUs) clustered at 100% identity threshold [32]. After the stringent quality control, 106,251 and 489,277 high-quality 16S rRNA gene or ITS2 amplicon sequences were obtained, respectively. High-quality 16S rRNA gene amplicon sequences ranged from 3236 to 10,625 sequences per sample (median reads per sample = 10,112), whereas the high quality ITS2 amplicon sequences ranged from 11,181 to 48,927 (median reads per sample = 24,686). In order to easily differentiate between bacterial and fungal ASVs, ASV names from the dataset were renamed as “bASV” or “fASV”, respectively, prior to further analysis. Once the ASV tables were obtained, they were converted into BIOM format [33] and imported into QIIME2 version 2019.10 [34] for downstream analysis as follows: A phylogenetic tree was built using q2-alignment [35] and q2-phylogeny [36] plugins. A pre-trained Naïve Bayes classifier based on SILVA SSU v138 [37] or UNITE v8.0 [38] databases, for bacterial and fungal datasets, respectively, was used for taxonomy assignment of the identified ASVs by using the q2-feature-classifier plugin [39]. Alpha- and beta-diversity were analyzed by using q2-diversity [40,41] and q2-taxa plugins [34]. Bacterial and fungal richness (Chao1 index, [42]), diversity (Shannon index, [43]) and evenness (Simpson index, [44]) metrics were calculated. For beta-diversity studies, bacterial and fungal samples were rarefied to 3236 and 11,181 reads per sample, respectively, in order to avoid biases due to different sequencing depths, and Bray-Curtis dissimilarity [45] distance matrices were calculated. Good coverage values and rarefaction plots showed that the bacterial diversity was sufficiently covered at the selected sampling depth for beta-diversity studies (Appendix A).

Pearson correlation between the relative abundances of 10 most abundant bacteria and fungi at the different ripening times and facilities was investigated by using the *cor* function in R environment v3.6.1 [31]. Plotting was carried out in R environment using *corrplot* v0.84 [46], *dplyr* v1.0.2 [47], *ggplot2* v3.3.0 [48], and *rehsape2* v1.4.3 [49] packages. Heatmap visualization of Appendix A was performed by using JColorGrid v1.860 [50]. The data for this study have been deposited in the European Nucleotide Archive (ENA) at EMBL-EBI under accession number PRJEB40652 (https://www.ebi.ac.uk/ena/browser/view/PRJEB40652).

## 3. Results

### 3.1. Quantification of Bacteria and Fungi in VB Rinds throughout Ripening Using qPCR and dPCR

Total abundance of bacteria and fungi on VB rinds (n = 200) from two different facilities throughout ripening (0, 14, 30, 90, 160 days of ripening) was evaluated by quantification of the 16S rRNA and 18S rRNA genes by using qPCR assays, respectively (Figure 1, Appendix A). Bacterial cell equivalents (BCE) per 0.5 g cheese rind were taken from our previous study [16] and included in the analysis. The results confirmed that fungi, in terms of fungal cell equivalents (FCEs), are abundant on VB rinds throughout ripening, even though their values were lower than bacterial cell equivalents (BCEs) (Figure 1). Overall, a higher abundance of fungi was found in facility A compared to facility B (Figure 1, Appendix A). At day 0, fungi were more abundant in facility A as compared to B (median FCE values of 2.20 ×$10^{8}$ and 7.52 × $10^{6}$, respectively). Inversely, bacteria were more abundant in facility A than B at 0 days of ripening, even though a higher abundance of total bacteria was found in facility B compared to facility A. Bacterial abundance from facility B increased at 14 days of ripening, whereas bacteria from facility A and fungi from facilities A and B abundance decreased significantly at this time (p<0.001). After this point, BCEs and FCEs values remained constant for facility A throughout ripening whereas for facility B, BCEs and FCEs increased significantly at 90 days of ripening and decreased again at 160 days of ripening (Figure 1, Appendix A).

Additionally, for the fungi quantification, FCEs were calculated and compared by using qPCR and dPCR targeting the 18S rRNA gene from the same 200 cheese rind gDNA samples already pooled according to their ripening time points and compared as described before. The comparative analysis of both assays on pooled cheese rind samples showed a strong linear correlation between qPCR and dPCR measurements, as stated by the coefficient of determination (R2) values (facility A: 0.998; facility B: 0.903). The negative controls for qPCR and dPCR did not show amplification. Examples of the dPCR results obtained, required for dMIQE (Appendix A), are represented in Appendix A. To study the effect of DNA digestion (outside as well as within the target) on the performance of the dPCR assay, quantification experiments were performed in parallel with non-digested and digested (EcoRI; AluI) aliquots of the same DNAs. We observed a slight improvement in generating positive droplet populations, resolution and rain by the application of EcoRI enzyme (Appendix A). By application of the internal (within the target) AluI enzyme, the target was mostly destroyed (Appendix A). However, since 10${}^{6}$–10${}^{7}$ target copies per μL template resulted in 100% saturation of positive droplets, the upper quantification limit for the dPCR was notably lower in comparison to qPCR. Therefore, the quantification of high levels of targets is a limitation of the dPCR compared to the qPCR assay. In general, the qPCR measurement (FCE/0.5 g) was 10 to 15 times higher when compared to the dPCR (Cp/0.5 g) (Appendix A). However, the biological importance of this finding is less meaningful. Additionally, gDNA (*S. cerevisiae* NCPF 3178) used for qPCR standard were verified by dPCR and indicated the same range of quantification as by qPCR (data not shown).

### 3.2. Investigation of Microbial Communities in VB Rinds throughout Ripening by Gene-Targeted Sequencing

#### 3.2.1. Bacteria

Overall, 106,251 high-quality near full-length 16S rRNA gene amplicon sequences were obtained (median = 10,112 per sample) and clustered into 134 ASVs, based on single-nucleotide differences (Appendix A). From these, 14 ASVs showed an overall relative abundance of over 0.1%, whereas only one ASV (bASV-01, assigned as *Staphylococcus equorum*) showed a relative abundance of over 1% overall and was present in all the samples studied. The 16S rRNA gene ASVs were classified into four bacterial phyla: *Firmicutes* (64.9% relative abundance overall), *Actinobacteria* (18.7%), *Proteobacteria* (15.9%) and *Bacteroidetes* (0.5%).

Figure 2a shows the relative abundance of the 10 most abundant genera overall. The most abundant genus was *Staphylococcus* (56.7% relative abundance overall), followed by *Brevibacterium* (9.9%), *Halomonas* (8.4%), *Psychrobacter* (7.4%), and *Corynebacterium* (4.9%). *Staphylococcus* was the most abundant genera in VB rinds at every ripening time point except at 90 days in facility A. The long read lengths provided by PacBio sequencing, coupled with the high fidelity resolution from the ASVs identification method, allowed us to deeply investigate the taxonomy of *Staphylococcus*: the genus was composed of 18 ASVs, 15 of which were assigned as *S. equorum* according to the SILVA database (the other 3 were assigned as *Staphylococcus* sp.) and accounted for more than 99.9% of all *Staphylococcus* 16S rRNA gene amplicon sequences (Appendix A).

The bacterial communities evolved differently throughout ripening in each facility (Figure 2a and Appendix A). The investigation of alpha-diversity metrics revealed that bacterial diversity in VB rinds from facility A increased throughout ripening time (Appendix A). This pattern was not observed for facility B, where the greatest Shannon value was observed at 14 days of ripening and the lowest at 160 days. The lowest bacterial richness and diversity (according to Chao1 and Shannon metrics) were observed in both facilities at ripening time 0. At this time point, VB rinds from both facilities were dominated by *Staphylococcus*, *Streptococcus* (formed by two ASV, both assigned as *S. salivarius* subsp. *thermophilus*), and *Lactobacillus* (formed by 11 ASVs, 4 of them assigned as *L. delbrueckii*). Both, *Streptococcus* and *Lactobacillus* were used as starter cultures for milk coagulation and were found mainly in VB rinds at 0 days of ripening. As ripening continued, several bacterial genera flourished: between 0 and 14 days *Psychrobacter* grew while decreasing after 30 days in both facilities. *Brevibacterium* [22 ASVs, four of them assigned as *B. linens* (that harbored 32% of all *Brevibacterium* 16S rRNA gene amplicon sequences) and three as *B. yomogidense* (12.4%)], *Halomonas* [31 ASVs, one of them assigned as *H. variabilis* (5.3% of all *Halomonas* 16S rRNA gene amplicon sequences) and one as *H. subglaciescola* (0.6%)] and *Corynebacterium* [6 ASV, 3 of them assigned as *C. casei* (16.0% of all *Corynebacterium* 16S rRNA gene amplicon sequences)] were abundant after 14 days of ripening and dominated with *Staphylococcus* at 160 days of ripening, especially in those VB rinds from facility A. *Leucobacter* (formed by 1 ASV), that appeared between 30 and 90 days of ripening in both facilities, and was also identified in the products at 160 days of ripening, especially in those from facility A. *Halomonas* was more abundant in facility B than in A and showed its greatest abundance at 14 days of ripening.

Bacterial beta-diversity in VB rinds based on Bray-Curtis dissimilarity distance matrices was evaluated and presented as a Principal Coordinates Analysis (PCoA) in Figure 3a. The differences between samples at early and late time points during the ripening process can be observed. Samples at the earlier and later ripening times from the ripening facility A separated more distant from each other than those front the facility B, revealing higher microbial dissimilarity. In facility A, VB rinds from earlier ripening times were dominated by *Staphylococcus* ASVs, whereas in the later time points *Brevibacterium*, *Corynebacterium* and *Leucobacter* appeared to be as abundant as *Staphylococcus*. In facility B, despite of the dynamics during ripening, *Staphylococcus* dominated at every time point and therefore samples cluster closer in Figure 3a.

#### 3.2.2. Fungi

The 489,277 high-quality ITS paired-end sequences obtained (median = 24,687 sequences per sample) were clustered into 69 ASVs that were taxonomically assigned to two different phyla, *Ascomycota* and *Basidiomycota*, although the latter one was found only in facility B at 0 days of ripening and at a very low relative abundance (0.5%) (Appendix A). Nineteen ASVs were found to be above an overall relative abundance of 0.1% and 3 ASVs, assigned as *Candida apicola* (ASV-1), *Debaryomyces hansenii* (ASV-2), and *Candida* (*Wickerhamiella*) *versatilis* (ASV-3), were over 1% (Appendix A).

Nine ASVs failed to obtain any taxonomy assignment below the kingdom level by using the UNITE database. Those “unassigned fungi” made up 7.8% of all ITS sequences. One of them, fASV-05 (best match in the UNITE database corresponding to “uncultured *Agaricales*”, MF484486, with 91% query coverage and 92% identity), was present in all VB rinds, except in those from 14 days after ripening in facility A, and harbored an overall relative abundance of 7.7% (corresponding to >99% of all the “unassigned Fungi” ITS2 sequences identified in this study).

The relative abundances of the ten most abundant fungal genera overall are displayed in Figure 2b. Similar fungal community dynamics occurred for both facilities throughout ripening. *Candida* was the most abundant fungal genus at time point 0 and its abundance decreased thereafter. *Debaryomyces* was the most abundant genus at 14 and 30 days of ripening in both facilities, followed by *Yamadazyma* and *Penicillium* in facility A, and *Scopulariopsis* in both facilities. *Scopulariopsis* was found in all VB rinds except in facility A at 0 days of ripening. The dynamics of *Scopulariopsis* followed similar evolution in both ripening facilities: it was absent or very low abundant during the first 14 days of ripening, then it increased towards 30 days, decreased towards 90 days and then increased again at 160 days of ripening, being the most abundant fungal genera in VB rinds from the latest time point investigated.

The fungal composition in VB rinds at 160 days of ripening differed between both facilities, despite of *Scopulariopsis* being the most abundant genera in both facilities. In facility A, VB rinds were dominated by *Scopulariopsis* (43.4% relative abundance), *Arachnomyces* (20.1%), *Fusarium* (11.3%), the unassigned fungi fASV-05 (8.6%), *Debaryomyces* (4.7%), *Aspergillus* (4.2%), and *Chrysosporium* (4.1%). In facility B, VB rinds were dominated by *Scopulariopsis* (52.5%), the unassigned fungi fASV-05 (37.7%), and *Candida* (6.5%). Fungal communities’ evolution in VB rinds during ripening can also be observed by analyzing the alpha-diversity indices, as Chao1 and Shannon values are greater in VB rinds from facility A at 160 days of ripening than in facility B (Appendix A). Even though *Candida* dominated at 0 days of ripening in both facilities, the Chao1 and Shannon metrics indicate that the species richness and diversity is higher in VB rinds from facility B than facility A. This can be explained by the different species within *Candida* that were found in VB rinds from each facility: in facility A, *Candida* was mostly represented by *C. apicola* (95.9% of all *Candida* sequences), whereas in facility B, *Candida* was represented by *C. versatilis* (40.9% of all *Candida* sequences), *C. tropicalis* (36.4%) and *C. apicola* (22.6%) (Appendix A).

The analysis of beta-diversity based on fungal diversity within samples was evaluated by Bray-Curtis dissimilarity distances and shown as a PCoA in Figure 3b. Fungal dynamics evolved similarly in both facilities, thus, samples are clustered according to the ripening time regardless of the facility they were sampled, with the samples from latest ripening times clearly separated from the earliest ones.

#### 3.2.3. Investigation of the Potential Correlation between Bacterial and Fungal Communities in VB Rinds throughout Ripening

Cheese ripening is a dynamic process where the microbes use the available nutrients for their growth, thereby producing secondary metabolites that other microbes can use, thereby continuing the pathway. With the aim of highlighting potential correlations between the different bacterial and fungal genera identified in VB rinds throughout ripening, Pearson correlation coefficients were calculated between the relative abundances of the 10 most abundant bacterial and fungal genera at the different ripening times and facilities and presented as a correlogram in Figure 4. Positive correlations were considered between those pairs of organisms that showed similar dynamics in both ripening facilities throughout the different ripening times (p< 0.05).

*Staphylococcus*, the most abundant bacterial genus overall, did not show positive correlations with any other bacterial or fungal genera. As it can be seen in Figure 2a, *Staphylococcus* was present in VB rinds at every time point and in both facilities, being also very abundant at the latest ripening times investigated (160 days). It was the only genera from this study showing such dynamics through time and thus no positive correlations were found with any other bacterial or fungal genera. Additionally, it was negatively correlated with *Alkalibacterium*, *Brachybacterium*, *Corynebacterium*, *Leucobacter* and *Fusarium*. Even though these bacteria and fungi were also detected in most VB rinds from both facilities at every ripening time, they showed a higher relative abundance in facility A at 90 and 160 days of ripening, where the lowest relative abundance of *Staphylococcus* was found in the study (Figure 2), thereby resulting in negative correlation with *Staphylococcus*. Alternatively, *Alkalibacterium*, *Brachybacterium*, *Corynebacterium*, *Leucobacter* and *Fusarium* showed positive correlations between all of them.

*Brevibacterium* was mainly present in VB rinds from the latest ripening points investigated and showed a positive correlation with *Corynebacterium* and *Scopulariopsis* that also were particularly abundant in those VB rinds. Additionally, *Brevibacterium* was negatively correlated with *Lactobacillus* and *Candida* which were highly abundant in VB during the first days of ripening (and decreasing thereafter), where the abundance of *Brevibacterium* was low. On the other hand, *Lactobacillus* and *Candida* were positively correlated with *Streptococcus* that was only found at 0 days of ripening in both facilities.

The fungal genus *Debaryomyces*, which was present in all VB rinds and was particularly more abundant in those after 14 and 30 days of ripening in both facilities, showed a positive correlation only with the bacterial genus *Psychrobacter*, that was also found mainly in VB rinds at 14 and 30 days. *Aspergillus* and *Penicillium*, which were mainly identified in VB rinds from facility A from 30 to 160 days of ripening, exhibited a positive correlation between them.

Other bacteria and fungi, despite of being identified in most VB rinds at different ripening points in both facilities, did not show any significant positive or negative correlation with other VB rind microbiota. That is especially the case for *Halomonas*, *Yamadazyma* and the fASV-05 (which did not get any taxonomic assignment in the UNITE database).

## 4. Discussion

The investigation of surface-ripened cheese rinds microbiota is pivotal due to its key role in the quality, safety, and organoleptic properties of the final products [3]. In this study, we focused on the characterization of the bacterial and fungal microbiota present in the rinds of Austrian Vorarlberger Bergkäse (VB), a surface-ripened cheese produced from raw-milk without the addition of external ripening cultures, throughout their ripening period by using culture-independent quantitative and qualitative approaches.

Bacterial quantification was performed by using 16S rRNA gene-targeted qPCR in a previous study done by our group [16]. In this study, we aimed to quantify fungi on VB rinds, and, by doing so, we evaluated two methodologies, qPCR and dPCR, both targeting the 18S rRNA gene, which has been previously recommended for fungal quantification [24,51,52,53]. qPCR has been widely used for fungal quantification in dairy products, as it overcomes biases related to cultivation-dependent methods [18,19,20]. However, its utilization is subjected to several biases, including target gene copy number variation, the selection of genetic markers, primers and the reference organisms required for performing the standard curve [23,54,55]. In this regard, dPCR has arisen as a more accurate methodology, as it provides more precise estimations of copy numbers and does not require comparison with external standards [21,22,23]. Therefore, we tested the potential of the dPCR for fungi quantification in VB cheese rinds, which has not yet been described. Regardless of the quantification method (qPCR, dPCR) used, our results showed that fungi were abundant in VB rinds throughout the ripening process. The comparison with bacterial quantification showed that FCE were 3–4 logs below BCE, which is in agreement with previous studies where bacterial counts have outnumbered the fungi [56,57,58,59]. Previous studies highlighted the greater efficiency of dPCR, especially when low numbers of target molecules occur in the samples [60]. This is not the case in cheese rinds, were culture-dependent studies (reviewed by Fröhlich-Wyder, et al. [59]) have shown that yeasts could reach 6–8 CFU/cm${}^{2}$ in cheese rinds during the first days of ripening. At higher concentrations, as also shown in the VB-cheese rind samples, the quantitative resolution of the dPCR is not given [61]. This is typically based on the method design and the range of concentrations [21,62]. Accordingly, the 18S rRNA gene dPCR assay used in this study determined less target sequence copies than expected by the calculated gDNA copy numbers. These discrepancies can be explained by the high concentration of the already diluted samples close to the limitation of the method design at approximately 25,000–30,000 droplets per chamber (Stilla Tech.). Nevertheless, our results demonstrated high correlation between qPCR and dPCR for the quantification of fungi in pooled VB cheese rind samples.

Despite the advantages previously described for dPCR [21,22,23], our results suggest qPCR targeting the 18S rRNA gene as a better cost- and time-effective analysis for the quantification of high-biomass VB-cheese rind samples.

The qualitative investigation of bacterial and fungal communities on VB rinds throughout ripening was conducted by using high-throughput gene-amplicon sequencing targeting the 16S rRNA gene and the ITS2, respectively. The ITS2 was chosen over the 18S rRNA gene (that was used for fungal quantification in this study) due to its higher taxonomic resolution [63]. Additionally, long-read PacBio sequencing was performed for bacterial investigation in order to reach the longest 16S rRNA gene sequence possible and therefore achieving deeper taxonomic resolutions.

The results showed dynamic changes of bacterial and fungal microbiota throughout ripening, with the succession of several bacteria and fungi strongly linked to specific ripening periods, regardless the facility where they were identified. VB products are sold at different ripening periods (usually 3, 6, 10, 12 and 18 months) that greatly influence their organoleptic properties and have a significant impact in the market. Therefore, investigating the bacterial and fungal that are present in VB rinds at specific ripening times and how they evolve through ripening might be greatly fruitful in order to deepen in our knowledge of VB production.

At 0 days of ripening, when the VB wheels are placed in the ripening room after the brine bath, we found rinds from both facilities being dominated by *Lactobacillus*, *Staphylococcus*, *Streptococcus* and *Candida*. Strains of *Lactobacillus* and *Streptococcus* were used as starter inoculums for milk coagulation and were mainly found in VB rinds at 0 days of ripening, so the presence of these two bacterial genera in VB rinds might be due to this artificial inoculation.

*Candida* is a yeast often found in raw cow milk, as well as other yeasts, such as *Kluyveromyces* and *Saccharomyces* [58,64,65,66]. We also identified *Kluyveromyces* and *Saccharomyces* in VB rinds on day 0 in both facilities, although with very low relative abundance. As we did not investigate milk microbiota, we cannot confirm milk as the source of these fungi. *Candida* has also been previously identified in the rinds of other European surface-ripened cheeses [8,57,67,68]. Some studies suggest that their presence on cheese is due to natural contamination from the environment [69]. *Candida* is a high-heterogeneous yeast genus and in our study was composed of 9 ASVs whose species-assignation and identification varied depending on the ripening facility. In facility A, *C. apicola* was the most abundant *Candida* species. It is an osmotolerant yeast with biotechnological use, especially in oil refineries, due to the enzymes with lipolytic activity and biosurfactants that it produced [70,71]. In facility B, the most abundant *Candida* species was *C. versatilis*, a high-salt tolerant yeast previously identified in brine and to be involved in miso and soy sauce fermentations [72,73]. As we did not investigate milk or brine in this study, we cannot ensure the provenance of such yeast on VB rinds. However, despite not being a starter culture, *Candida* was highly abundant at 0 days of ripening and might impact VB ripening, as it was also identified at all the ripening stages investigated.

Another organism that was identified as a non-inoculated, “first colonizer” in VB rinds was *Staphylococcus*. It was the most abundant bacterial genus in VB rinds at 0 days of ripening and was also very abundant in all the other time points in both facilities, including the latest time points investigated (160 days). Almost all (>99.9%) *Staphylococcus* 16S rRNA gene amplicon sequences were assigned to *S. equorum*. The source of this bacteria in VB rinds might be the natural inoculation from the processing environment, as previous studies performed by our group identified *S. equorum* to be abundant in VB rinds and also in many surfaces from the ripening cellars, including the shelves and racks where the VB wheels are placed, floor, walls and air filters [14,15]. Furthermore, *S. equorum* has been associated with important ripening functions, such as lipolytic and proteolytic activity and the production of volatile compounds and antimicrobials [8,74,75,76]. This species appears to be pivotal for the production of VB, although its specific contribution to VB ripening and its organoleptic properties must be further investigated.

As soon as the ripening process began, we identified the yeast *Debaryomyces* to be highly abundant at days 14 and 30 in both facilities and decreasing thereafter. This is in agreement with former studies that identified *Debaryomyces* in young cheeses, where they used the lactic acid produced by the starter cultures after lactose degradation [7,8,59,75,77]. In our study, *Debaryomyces* was positively correlated with *Psychrobacter*, as they were both very abundant in rinds of young VB cheeses (14–30 days of ripening) in both facilities. The whole-genome analysis of a *Psychrobacter* strain (L7), previously isolated from VB rinds by our group [16], revealed enzymatic features related to lipolysis, proteolysis and the production of aroma compounds. Therefore, the presence of *Psychrobacter* and *Debaryomyces* as “early colonizers” in VB rinds at the earlier ripening stages might be due to their capability of using primary metabolic compounds present in the young cheeses and those released by the “first colonizers”: *S. equorum*, *Candida* and the starter cultures. The origin of *Psychrobacter* might be the production environment, as it was previously identified on different surfaces of the VB ripening cellars [15].

At the longest ripening times (160 days), we identified high abundance and significant positive correlation of *Brevibacterium* and *Scopulariopsis* in both facilities. *Brevibacterium* is usually found in surface-ripened cheeses, in VB rinds and in the production environment [7,9,12,14,15,65,78]. Most *Brevibacterium* 16S rRNA gene sequences were classified as *B. linens*, which is appreciated for the production of carotenoids and sulfur-containing compounds, which are enhancing cheese organoleptic properties [79]. *Scopulariopsis* has also been commonly identified in long-ripened cheeses, included VB [14,80]. It has been stated that during the early ripening times, filamentous molds such as *Scopulariopsis*, slowly generate their hyphae network to access the nutrients of the cheese matrix while they reach an exponential growth phase later, therefore being identified more often in long-ripened cheeses [56,64]. Kastman, et al. [80] reported that *Scopulariopsis* produces siderophores enhancing *S. equorum* growth. However, we did not find a positive correlation between these two organisms. As stated in the Results section, *S. equorum* harbored unique dynamics in VB rinds in this study, being the only organism to be highly present at every ripening time, from the first to the last one, and in both ripening facilities. The correlation analysis was performed to highlight potential correlations between pairs of organisms that showed similar dynamics throughout the entire process and in both facilities. Therefore, as there was no other organism from this study with the same dynamics as *S. equorum*, this bacteria did not show positive correlation with any other bacterial or fungal genera. However, this does not necessarily mean that *S. equorum* is not benefitting from other bacterial or fungal species growth or their related metabolites. The specific relationships between *S. equorum* and the other bacteria and fungi identified in VB rinds in this study will require further investigation.

Additionally, other microorganisms were found to be abundant in VB rinds at the latest ripening times (90 and 160 days), such as *Corynebacterium*, *Leucobacter* and *Fusarium*, that were also positively correlated between them. *Corynebacterium* and *Leucobacter* are commonly identified in surface-ripened cheeses, where they contribute to lipolytic and proteolytic activities and to the production of appreciated color and aroma compounds [9,12,68,81,82,83]. Both have also been previously identified in VB rinds and in the producing environment [14,15]. Even though several species of *Fusarium* have been related to cheese defects, the ASVs identified here were assigned to *F. domesticum*, which was isolated from the surfaces of various European cheeses and has been characterized for its ability to increase the drying of the cheese rinds, thus reducing its stickiness [84]. The bacteria *Brevibacterium*, *Corynebacterium* and *Leucobacter*, and the fungi *Fusarium* and *Scopulariopsis* can be considered as “late colonizers” in VB rinds, and their potential contribution and cooperation for the organoleptic properties of this products must be addressed in the future.

Some bacteria and fungi were present in a relatively high abundance in most VB rinds throughout ripening but did not show any positive or negative correlation with any other organism in this study. That was the case for *Halomonas* and fungus *Yamadazyma*. Both genera are represented by highly halotolerant species. All *Yamadazyma* ASVs in this study were classified as *Y. triangularis*, a yeast previously found in brine, food processing environments, in cheese and on the surface of dry-cured ham [85,86,87]. *Halomonas* is an heterogeneous genus ubiquitously identified in saline or hypersaline environments and also in milk and cheeses, including VB rinds [14,15,65]. In our previous study [15], we identified *Halomonas* as the most abundant genus on the surfaces present in ripening cellars, although the species abundance in cheeses and/or on surfaces varied. In this study, *Halomonas* was composed of 31 ASVs, being the genus composed of the greatest number of ASVs overall, supporting the high heterogeneity of the genus identified previously in VB producing environments. The role of *Halomonas* and *Yamadazyma* on cheese ripening is unclear. However, due to the high salt concentration during VB production and the fact that they were present and abundant at many different time points with no specific dynamics pattern or correlation to the dynamics of other bacteria and fungi, it can be suggested that *Halomonas* and *Yamadazyma* spuriously colonize the surface of the product from the environment and flourish on the surface of VB by being favored by their resistance to the salt conditions. Further studies including gene-expression investigation (via metatranscriptomics, etc.) will reveal the real impact of these organisms on VB ripening.

Another organism that did not show any correlation with other bacteria or fungi was the fungal fASV-05. This ASV did not reveal any taxonomic assignation according to the UNITE database, whose best match only reached order level (“uncultured *Agaricales*”), although the sequence alignment values were very low to be considered a real match (91% query cover and 92% identity). Further investigation, including whole-DNA shotgun sequencing, and/or culture-dependent methods, might be performed to deeper investigate this organism, as it was found to be very abundant in VB rinds (particularly at day 90). The lack of information in the current databases might suggest that this represents a potential novel fungal lineage/phylotype, which should be verified by the usage of additional or alternative markers like the β-tubulin (*tub2*), the translation elongation factor 1-α (*tef1*α) or the second largest subunit of ribosomal polymerase II (*rpb2*) in further studies [88,89,90].

The results obtained in this study expand our previous knowledge of the microbial communities involved in VB ripening. Even though we focused on this high-value regional product, the results and methodology described can be extended to other cheese products, as most of the microbiota described here have been previously reported to have a significant impact in the safety and organoleptic properties of other cheeses [2,3,4,5]. qPCR and gene-targeted amplicon HTS have arisen as powerful, rapid and cost-effective methods for the quantification and characterization of the microbial communities occurring in cheeses and has been applied worldwide. This methodology can be used for different traditionally manufactured cheeses, regardless of the country or the manufacturing procedure. The effect of the facility-specific microbiota in shaping the microbial communities of the product is pivotal [6,7,15]. The environmental microbiota is able to naturally colonize and flourish on the surface of the cheese and is influenced by the physicochemical conditions occurring either on the product (e.g., pH, water activity, nutrient sources) or in the ripening cellar (e.g., temperature, humidity, salt concentration, frequency of washings). Non-inoculated bacteria and fungi dominated in VB rinds throughout ripening and were previously identified to dominate different surfaces from the ripening cellar by our group [15]. As not all factors acting as potential vectors for microorganisms or influencing their development have been addressed, the presented results of the cheese rind microbiota should be considered as microbiologically linked ecosystems between the cheese and the processing environment.

The successional evolution and correlation of the VB rind microbiota throughout ripening leads the way for further investigation towards a more standardized and safer manufacture of VB while enhancing the organoleptic properties. The ripening time is a key factor, as it widely influences the aroma, flavor and texture of VB products and has an impact on the consumer buying behaviour. Despite of different conditions (temperature, humidity, salt concentration, dry salting, and frequency of the salt baths) that might have influenced the microbial composition of the products, we described similar successional events occurring in the abundance of the microbial communities in both facilities. Considering the possibility of customization of these physicochemical characteristics, further investigation involving greater number of facilities and ripening conditions must be conducted in order to highlight factors that might favor the growth of key bacteria and fungi.

The microbial composition of VB cheese and the processing environment [14,15] as well as the genomic features of some of the more abundant bacteria [16,17] were previously described by our research group. Here, we used gene-amplicon high-throughput sequencing approaches, which allowed us to investigate the microbial communities deeper than in our former studies conducted by using cloning and Sanger sequencing. We are aware of the limitations and biases related to gene-targeted amplicon sequencing, in terms of differential gene copy number, preferential primer linkage, biases during DNA extraction, library preparation and discrimination between live and dead cells. However, the high number of samples used here, in combination with our previous studies, highlights several bacteria and fungi to be consistently dominant in rinds during VB ripening. These bacteria and fungi must be the target of further studies involving metagenomic and metatranscriptomic shotgun sequencing to unravel the entire genomic content and gene expression profile. It is important to highlight the gene expression profiles of those metabolic pathways that are essential at the beginning of the cheese ripening (e.g., proteolysis, lipolysis) or the generation of organoleptic and volatile compounds (e.g., metabolism of amino acids, fatty acids). Furthermore, these approaches must be combined with chemical analyses and sensorial evaluations at different stages of the VB cheese manufacturing process and under different ripening conditions, to improve its production in terms of the organoleptic properties and safety.

## 5. Conclusions

Bacteria are more abundant than fungi in VB throughout ripening, although both kingdoms were abundant throughout the process. The microbiota of VB rinds evolved throughout ripening, and different bacteria and fungi were identified in both facilities at specific ripening periods, regardless of the ripening facility investigated. The results point out several microorganisms, such as *Staphylococcus*, *Brevibacterium*, *Corynebacterium*, *Leucobacter*, *Scopulariopsis*, *Debaromyces*, and *Candida*, to be highly abundant in the process. The correlation analyses suggest a strong linkage between certain bacteria and fungi and specific ripening periods, which might be relevant for the proper development of VB ripening. Further studies targeting the organisms identified here will deepen our understanding of surface-ripened cheese ripening and the potential functions of the rind microbiota.

## Figures and Tables

**Figure 1 foods-09-01851-f001:**
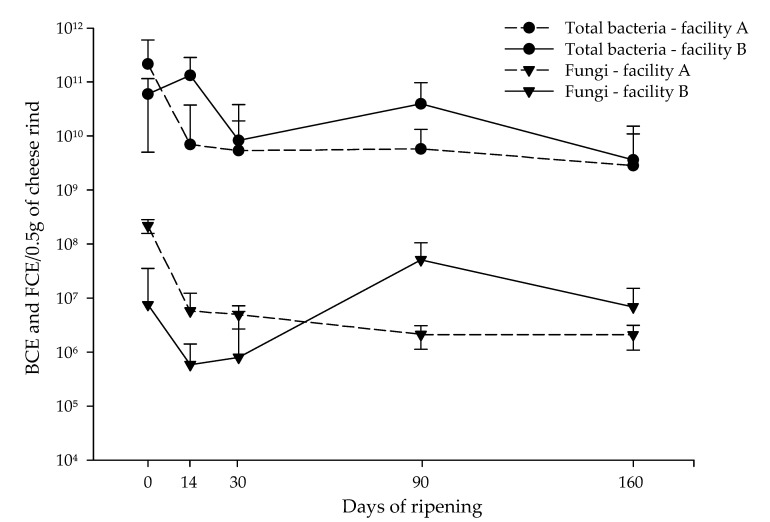
Abundance of cheese rind bacteria and fungi during ripening of Vorarlberger Bergkäse in two different cheese production facilities determined by qPCR. Bacterial cell equivalents (BCE) and fungal cell equivalents (FCEs) per 0.5 g cheese rind during ripening in two different cheese production facilities are shown. The graph shows median and interquartile ranges for the 20 samples from each analyzed day of ripening (0, 14, 30, 90, and 160 days) for each facility (A, B). The data for total bacteria count (BCE) was taken from Schmitz-Esser, et al. [16]. Numerical FCE values are shown in Appendix A, *p*-values are shown in Appendix A. Detailed information about 18S rRNA gene qPCR assay are listed in Appendix A (MIQE guidelines qPCR).

**Figure 2 foods-09-01851-f002:**
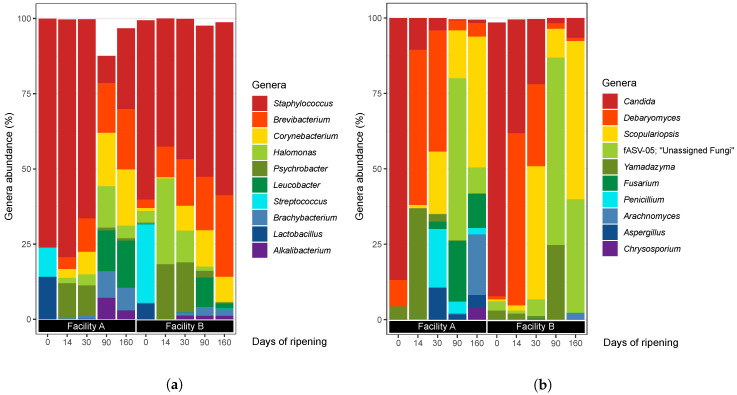
Relative abundance (*y*-axis) of the 10 most abundant bacterial (**a**) or fungal (**b**) genera in VB rinds, obtained by 16SrRNA gene or ITS2 amplicon high-throughput sequencing, respectively. The *x*-axis corresponds to the different sets of pooled samples taken from the two ripening plants and at the 5 different ripening times (20 samples per plant and ripening time).

**Figure 3 foods-09-01851-f003:**
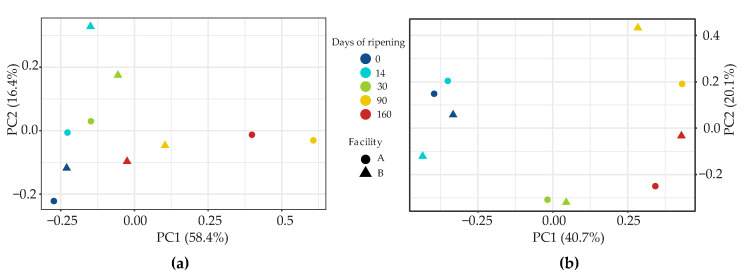
Principal Coordinates Analysis (PCoA) of the 16S rRNA gene or ITS2 Amplicon Sequence Variants (ASVs) based on Bray-Curtis distances for bacterial (**a**) or fungal (**b**) beta-diversity investigation, respectively. Samples are colored according to the ripening day. The shape of the symbols corresponds to the ripening facility (A, circles; B, triangles).

**Figure 4 foods-09-01851-f004:**
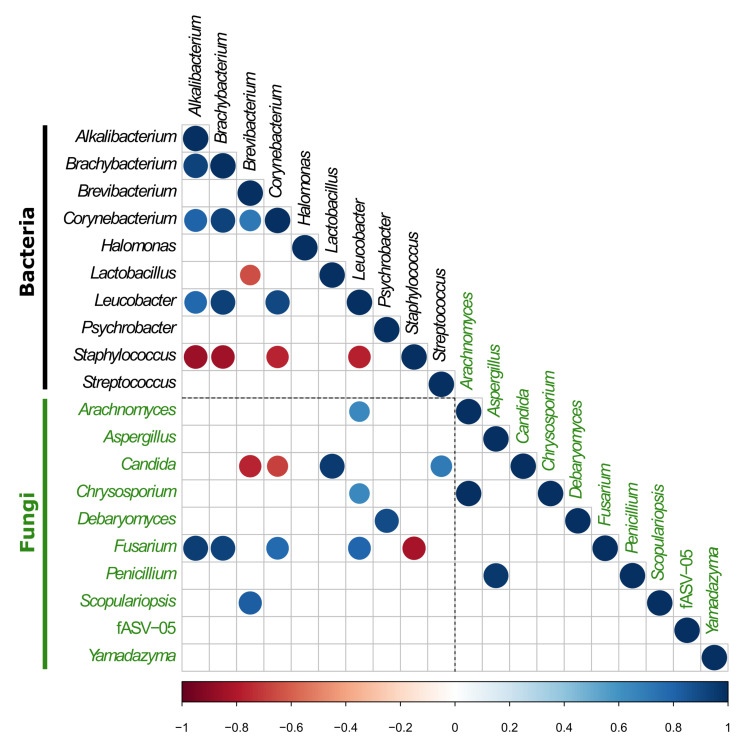
Pearson correlation analysis between the relative abundances of bacterial and fungal genera (obtained after gene-targeted high-throughput sequencing) at the different ripening times and facilities. Only statistically significant correlations are displayed (p< 0.05). Correlation coefficients are colored from dark red (negative correlation) to dark blue (positive correlation). Color intensity and the size of the circle are proportional to the correlation coefficients. Bacterial and fungal genera names are colored in black and green, respectively.

**Table 1 foods-09-01851-t001:** Ripening conditions according to the different VB producing facilities and the different time points investigated in this study.

Facility	Days of	Temperature	Humidity	Cheese Treatment	Brine	Dry
	Ripening	(${}^{\circ }$C)	(%)	with Brine	Concentration (%)	Salting *
A	0, 14, 30, 90	13.5	96	daily	20	no
	160	10	95–96	once a week	10	
B	0, 14, 30, 90	13	93–94	2–3 times a week	10	yes
	160	13	93–94	once a week	15	

* Dry salting of cheese surfaces was only applied in the facility B (during day 0 to 6; each side three times with 45 to 50 g NaCl per side). During this initial ripening period, the intervals of cheese treatments were reduced from daily washing procedures to washing two to four times per week at the same time as the brine concentration is reduced to 10% or less. Starter cultures (*Streptococcus thermophilus*, *Lactococcus delbrueckii* spp. *lactis*, *Lactobacillus casei*) were obtained from the Federal Institute for Alpine Dairying, Rotholz, Austria.

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
