# Peer review of "Austrian Raw-Milk Hard-Cheese Ripening Involves Successional Dynamics of Non-Inoculated Bacteria and Fungi"

_foods, 2020, doi:10.3390/foods9121851_

Round 1

Reviewer 1 Report

In my opinion the subject of the manuscript is very interesting and well described.

The introduction is well edited and written, I have no objections to this part of the manuscript.

I have comments for the rest of the work. As reported by the authors of the manuscript, Vorarlberger Bergkäse (VB) is an raw milk brine-washed hard-cheese, therefore, in my opinion, the discussion of the results should refer to the technology and microflora of hard cheese, salted in brine. I find it very useful to compare the current microflora depending on the ripening temperature of the cheeses, the method and parameters of salting the cheeses. Also he analysis between the relative abundances of bacterial and fungal genera (Fig. 4) should take into account the division of the present microflora considered to be technical and contaminating.

Author Response

Authors: We are very grateful for the reviews provided by the editor and each of the external reviewers. Please find our detailed response to the comments below. Line numbers refer to the marked-up version of the manuscript. Our response to the reviewers’ comments, replacements, deletions and additions are indicated as blue text. A list of the changes has been provided additionally at the end of the manuscript (L907-921).

#Reviewer 1: In my opinion the subject of the manuscript is very interesting and well described.

The introduction is well edited and written, I have no objections to this part of the manuscript.

I have comments for the rest of the work. As reported by the authors of the manuscript, Vorarlberger Bergkäse (VB) is an raw milk brine-washed hard-cheese, therefore, in my opinion, the discussion of the results should refer to the technology and microflora of hard cheese, salted in brine. I find it very useful to compare the current microflora depending on the ripening temperature of the cheeses, the method and parameters of salting the cheeses. Also he analysis between the relative abundances of bacterial and fungal genera (Fig. 4) should take into account the division of the present microflora considered to be technical and contaminating.

#Authors’ response:  We agree with the reviewer. The physicochemical conditions occurring during cheese ripening, either being inherent to the product (pH, water activity, nutrient availability, etc.) or to the ripening cellar (temperature, humidity, salt, etc.) are main aspects, as they influence the microbial communities present on cheese rinds. We are aware of the limitation of the study, as the environmental and physiochemical sources that can be subjected to human customization and their function in shaping the microbial communities were not addressed. Further studies on this type of cheese, including a greater number of facilities under different ripening conditions will be required in order to define specific ripening conditions that favor the growth of specific bacteria and could be used towards a safer manufacture while enhancing the organoleptic properties of the product.

The reviewer is correct, the food manufacturing environment plays a significant role in defining the final microbiota of the hard cheeses. It has been recognized as a source of natural microbial inoculation in one of our previous studies on VB (Quijada et al., 2018, DOI: 10.1016/j.ijfoodmicro.2017.12.025). We identified bacteria dominating in VB rinds to be present on different surfaces from the ripening cellar (including air filter, floor, shelves, racks and walls). The intention of Figure 4 was to give a descriptive overview of the potential correlation of species throughout ripening in VB rinds. From the microbiota described in the Figure 4, only Lactobacillus and Streptococcus were inoculated as starter cultures.

We strongly appreciate the reviewer’s comments and recognize their potential to improve the quality of the manuscript. Therefore, we added a new paragraph to the Discussion section (L541-L550 & L555-563) to address the comments.

Reviewer 2 Report

The authors investigated both bacterial and fungal communities from VB rinds (artisanal raw milk brine-washed hard-cheese manufactured in the western part of Austria) at different times throughout ripening (from 0 to 160 days of ripening) by both quantitative and qualitative approaches. 

The article is quite interesting, very good sought, the analytical part is very good and the presentation of results clear. It is also interesting that the authors present new data and have also future work to do as they have written. I believe that these types of works are very attractive to the research community and even though have more local interest are very helpful for the respective researchers.

The only suggestions that I have to make are the followings:

The authors should underline the efficiency of the proposed microbiological analysis in other also traditional cheeses of other countries.

In addition, a preliminary sensory evaluation or some other chemical analysis (proteolysis, volatile compounds) should be added even as future work in the manuscript. The microbiological population should be further investigated in combination with the organoleptic properties of the produced cheese during the ripening period and various parameters should be efficiently correlated.

Author Response

Authors: We are very grateful for the reviews provided by the editor and each of the external reviewers. Please find our detailed response to the comments below. Line numbers refer to the marked-up version of the manuscript. Our response to the reviewers’ comments, replacements, deletions and additions are indicated as blue text. A list of the changes has been provided additionally at the end of the manuscript (L907-921).

#Reviewer 2: The authors investigated both bacterial and fungal communities from VB rinds (artisanal raw milk brine-washed hard-cheese manufactured in the western part of Austria) at different times throughout ripening (from 0 to 160 days of ripening) by both quantitative and qualitative approaches.

The article is quite interesting, very good sought, the analytical part is very good and the presentation of results clear. It is also interesting that the authors present new data and have also future work to do as they have written. I believe that these types of works are very attractive to the research community and even though have more local interest are very helpful for the respective researchers.

The only suggestions that I have to make are the followings:

The authors should underline the efficiency of the proposed microbiological analysis in other also traditional cheeses of other countries.

#Authors’ response: We thank the reviewer for this important comment. The quantitative and qualitative methodologies used in this study represent a rapid and cost-effective approach for the in-depth characterization of the microbial communities from a certain environment and it could be applied to different types of cheeses, regardless of the country or the manufacturing procedure. Furthermore, the application of culture-independent methods in the investigation of the microbiota of traditionally manufactured cheeses will allow the identification of microorganisms unable to grow in vitro by using conventional laboratory techniques. We modified the Discussion section (L535-L541) accordingly in order to include the reviewer’s comment.

In addition, a preliminary sensory evaluation or some other chemical analysis (proteolysis, volatile compounds) should be added even as future work in the manuscript. The microbiological population should be further investigated in combination with the organoleptic properties of the produced cheese during the ripening period and various parameters should be efficiently correlated.

#Author’s response: We appreciate the reviewer’s comment and agree that describing the sensory properties of the cheese is potentially important. It is crucial to analyze the flavor and its correlation with the formulation and the cheese-making process as well as its interrelation with the rheological, textural and microstructural properties. The correlation of chemical analysis (concentration of volatile compounds, etc.) with sensorial evaluation (blind testing) and deeper shotgun DNA sequencing (metatranscriptomic analysis of the pathways involved in the generation of volatile compounds: metabolism of amino acids, fatty acids, etc.) will be pivotal steps in future research on this particular product and could be extended to other traditionally manufactured cheeses. We modified the Discussion section (L569-L577) to include this reviewer comment as well.